# MTVNet: Multi-Contextual Transformers for Volumes − Network for Super-Resolution with Long-Range Interactions

August Leander Høeg[*1], Sophia W. Bardenfleth[1], Hans Martin Kjer[1], Tim B. Dyrby[1,2], Vedrana Andersen Dahl[1], and Anders Dahl[1]

[1]Department of Applied Mathematics and Computer Science, Technical University of Denmark (DTU)
[2]Danish Research Centre for Magnetic Resonance (DRCMR), Copenhagen University Hospital Hvidovre
`aulho@dtu.dk`

## Abstract

Recent advances in transformer-based models have led to significant improvements in 2D image super-resolution. However, leveraging these advances for volumetric super-resolution remains challenging due to the high memory demands of self-attention mechanisms in 3D volumes, which severely limit the receptive field. As a result, long-range interactions, one of the key strengths of transformers, are underutilized in 3D super-resolution. To investigate this, we propose MTVNet, a volumetric transformer model that leverages information from expanded contextual regions at multiple resolution scales. Here, coarse resolution information from boarder context regions is carried on to inform the super-resolution prediction of a smaller area. Using transformer layers at each resolution, our coarse-to-fine modeling limits the number of tokens at each scale and enables attention over larger regions than previously possible. We compare our method, MTVNet, against state-of-the-art models on five 3D datasets. Our results show that expanding the receptive field of transformer-based methods yields significant performance gains on high-resolution 3D data. While CNNs outperform transformers on low-resolution data, transformer-based methods excel on high-resolution volumes with exploitable long-range dependencies, with our MTVNet achieving state-of-the-art performance. Our code is available at https://github.com/AugustHoeg/MTVNet.

## 1  Introduction

In recent years, super-resolution (SR) and other vision tasks have seen significant improvements via usage of vision transformers (ViTs). Although ViTs achieve state-of-the-art performance in 2D SR [1–4], few studies have attempted applying ViTs for volumetric SR. Part of the success of ViTs is their increased receptive field compared to Convolutional Neural Networks (CNNs), enabling inferences based on broader image context [5]. Based on experiences from 2D SR, it is logical to assume ViTs will out-

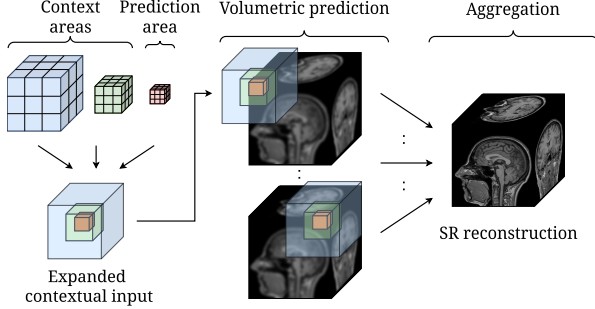

**Figure 1.** Overview of MTVNet that is informed by a large contextual volume processed at multiple resolution scales for predicting SR in the center volume.

perform CNNs in the domain of 3D data. However, in volumetric SR, ViTs are challenged by the cubic growth in tokens required to process larger 3D image contexts. Although window-based attention allieviates the quadratic complexity of self-attention [6], the complexity of 3D data still limits the receptive field of volumetric SR models. Because of this disadvantage, the performance gap of CNNs vs. ViT-based architectures for volumetric SR has yet to be fully understood.

Several works have studied enhancement of 3D medical data such as MRI (magnetic resonance imaging) and CT (computed tomography) by upscaling slices independently [7–11]. While such methods circumvent the complexity issues of volumetric SR, not fully considering the 3D context reduces performance and risks slice discontinuities [12–16].

Current brain MRI benchmark datasets for evaluating volumetric SR are relatively low-resolution [17], limiting the benefits of a larger receptive field. Advancements in medical imaging technology enable higher spatial resolution [18], resulting in larger volumes where volumetric SR can benefit from long-range contextual information. Given the potential of SR in clinical settings and the increasing interest in applications like multi-resolution synchrotron imaging [19], there is a need for volumetric SR methods designed for high-resolution (HR) 3D data.

Aside increasing contextual information in volumetric SR, recent studies in 2D SR have shown that the window-based attention mechanism of the

---

[*]Corresponding Author.

Proceedings of the 7th Northern Lights Deep Learning Conference (NLDL), PMLR 307, 2026.

Swin-Transformer [6] is not ideal for capturing relationships across distant image regions. Using Local Attribution Mapping (LAM), Chen et al. [2] showed that strengthening long-range information exchange can lead to significant performance gains. Similarly, recent ViT research has focused on modeling long-range interactions to increase performance [20, 21].

To address the limitations caused by self-attention, we present MTVNet, a volumetric SR approach that leverages multi-contextual information from regions beyond the prediction area (see fig. 1) and employs hierarchical attention to enhance long-range information propagation. This approach builds on the assumption that regions closest to the prediction area provide the most critical contextual information, while more distant regions contribute less. Consequently, we design a coarse-to-fine feature extraction and image tokenization scheme that allocates less compute to regions further from the prediction area, enabling larger volumetric inputs without exceeding GPU memory. Inspired by FasterViT [21], we introduce a hierarchical attention mechanism for volumetric image processing, improving modelling of long-range interactions to enhance SR performance.

Finally, MTVNet enables us to investigate the performance gap between CNNs and ViT-based methods for volumetric SR. We compare MTVNet with convolutional and ViT-based SR methods in both 2D and 3D across low-resolution brain MRI data and high-resolution CT data. Extensive experiments show that on low-resolution MRI datasets, CNNs outperform ViTs due to their stronger ability to model local image dependencies. This suggests that the architectural advantages of ViTs only emerge in high-resolution data, where long-range contextual information becomes more important. Conversely, in high-resolution data, we find ViT-based methods achieve superior performance, with our MTVNet leveraging broader contextual input to achieve state-of-the-art performance. These findings highlight that the relative performance of ViT-based methods in 3D highly depends on input resolution.

## 2 Related Work

### 2.1 Learning-based super-resolution

The benefits of learning-based SR over classical interpolation were first shown by SRCNN [22]. Several CNN-based models have since been proposed to improve performance and efficiency [23–27].

Despite the success of CNNs, many vision tasks, including image classification [5, 6, 20, 21], object detection [28–31], and segmentation [32–36] have seen improvements using vision transformers. In 2D SR, SwinIR [1] demonstrated the potential of ViTs over CNN-based models by incorporating the Swin Transformer [6] in a residual network scheme.

Building upon the success of SwinIR, Chen et al. [2, 37] proposed cross attention of overlapping window partitions and channel attention mechanisms to enable activation of more input pixels. Recently, Hsu et al. [4] suggested combining Swin transformer layers and gating mechanisms in a densely-connected structure [38, 39] to alleviate information bottlenecks. Although these methods achieve state-of-the-art performance in 2D SR, the increased complexity of 3D data makes them difficult to transfer directly to 3D, except when applied slice-wise.

### 2.2 Super-resolution for 3D volumes

Super-resolution of 3D volumes finds motivation in clinical applications, where workflows are highly dependent on the interpretation of fine-grained structures that are often undersampled during routine acquisitions. SR enhancement of these structures enables improved diagnostic sensitivity and treatment planning through more precise delineation of organs and lesions. SR reconstructions from LR scans allow shorter acquisition times and alleviate requirements for scanner hardware replacement, enabling increased scanner throughput and accessibility. In CT, SR reduces patient health risks by allowing lower radiation scan protocols without compromising image quality [12, 13, 15–17]. Recognizing these benefits, several 3D SR methods have been proposed, including slice-wise and volumetric approaches.

Slice-wise methods predict each slice independently, enabling support for deeper architectures but neglecting cross-slice information, potentially causing discontinuities in slice predictions. Volumetric SR methods fully utilize the context in 3D, increasing computational complexity but enabling better performance thanks to improved inter-plane modelling [12–16]. Inspired by SRCNN [22] and SRGAN [13], Pham et al. [40] and Chen et al. [13] proposed volumetric adaptations of convolutional SR models and demonstrated the potential of volumetric SR over slice-wise approaches. Research in volumetric SR has since grown rapidly and several methods have been proposed to improve efficiency and performance [12, 14, 15, 41–46]. These approaches are similar to 2D SR, only they aim to improve the image quality along all dimensions of a volumetric image instead. However, other approaches including axial SR models [47–49] have been proposed to increase the slice count of low-resolution MRI volumes while preserving in-plane resolution. To alleviate the limitation of fixed upscaling factors, arbitrary scale SR based on Implicit Neural Representation [45, 50, 51] have been proposed. Multi-contrast volumetric models [49, 51] that leverage information from multiple MRI modalities (T1- and T2-weighted images) have also been proposed. Recent advances in ViTs have also inspired volumetric SR methods.

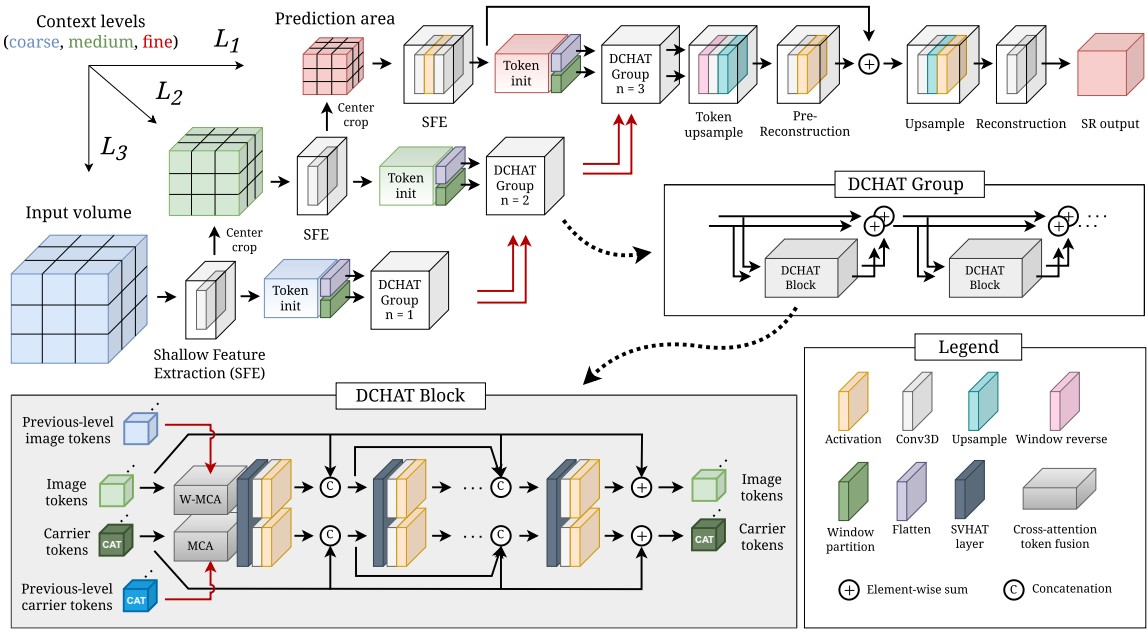

**Figure 2.** Illustration of MTVNet and the structure of DCHAT block and DCHAT group. Our proposed architecture consists of up to three network stages $L_1, L_2, L_3$ of multi-contextual volumetric image processing. In each succeeding network stage of MTVNet, we tokenize spatial subsets of the input volume using progressively smaller 3D patch sizes, resulting in both coarse- and fine-grained feature extraction. The depth of subsequent DCHAT groups increases from $n = 1$ to 3 DCHAT blocks towards the last network stage, which produces the SR prediction. Image tokens and carrier tokens from preceding network stages (red arrows) are fused into later stages using multi-head cross-attention (MCA) and window-based multi-head cross-attention (W-MCA).

SuperFormer [16] merged feature embeddings and volume embeddings using a volumetric transformer-based network structure similar to SwinIR [1]. Also inspired by SwinIR, Ji et al. [49] implemented a transformer-based GAN (generative adversarial network) for axial SR using residual swin transformer blocks [1, 6]. The CFTN model [52] used 3D residual channel attention blocks [26] and transformers to capture global cross-scale dependencies between multi-scale feature embeddings. Li et al. [51] proposed a 2D slice-wise multi-modal arbitrary scale SR model featuring a rectangle-window cross-attention transformer to model long-range dependencies.

### 2.3 ViT enhancements

With the increasing usage of ViTs across image tasks, several works have sought to address the scalability of self-attention. Liu et al. [53] proposed SwinV2, featuring improved normalization and a more robust attention mechanism using cosine similarly. EfficientFormer [54] proposed a lightweight ViT architecture featuring efficient attention mechanisms to achieve competitive accuracy and inference speeds. In CrossViT [20], multi-scale tokenization and efficient cross-attention mechanisms were used to extract and fuse feature representations at different image scales. In connection with scaling ViTs to higher input resolution, several works have suggested augmenting local attention to improve long-range interactions while maintaining efficiency. Twins [55] combined local attention and globally sub-sampled attention to improve efficiency and capture both local and long-range dependencies. RegionViT [56] suggested combining attention between local and regional tokens for conveying global information between attention windows, improving long-range interaction and efficiency. Similarly, FasterViT [21] proposed a hybrid CNN/ViT architecture featuring hiearchical attention mechanisms using local tokens and specialized carrier tokens. These works find natural applicability for volumetric image tasks due to the high data complexity. For instance, FINE [57] used global attention using memory tokens for improved 3D segmentation performance. Yet, to our knowledge, MTVNet is the first to leverage these concepts for volumetric SR.

## 3 Methods

### 3.1 Network architecture

The architecture of MTVNet consists of up to three network stages $L_1, L_2, L_3$ marked by respectively red, green and blue in fig. 2. Stages $L_3$ and $L_2$ extract features from regions beyond the SR prediction area and merges them into $L_1$. These features serve as a prior for stage $L_1$, producing a SR output conditioned on the surrounding image context.

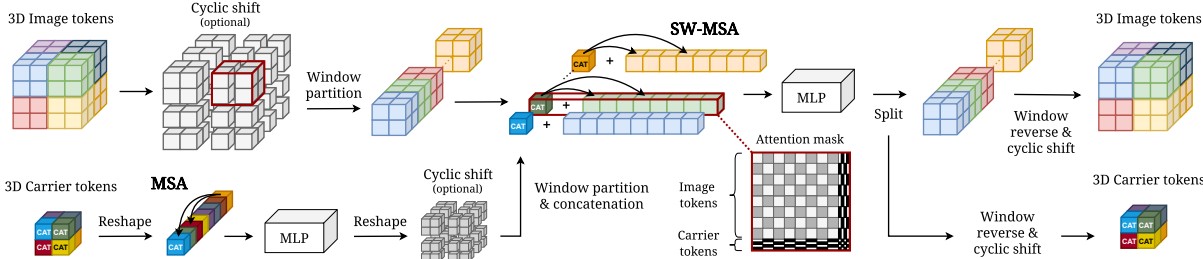

**Figure 3.** Illustration of our proposed SVHAT. Carrier tokens first undergo full self-attention (MSA) before being concatenated with the tokens of their respective local attention window. Shifted windowed self-attention (SW-MSA) is then performed within each attention window, with the carrier tokens enabling global information exchange between neighboring windows. Attention masking is used to drop information exchange between non-adjacent image tokens and carrier tokens, where grey and black areas indicate masked regions.

**Shallow Feature Extraction**. Each network stage performs shallow feature extraction (SFE) using $3 \times 3 \times 3$ convolutional layers, producing shallow feature embeddings $\mathcal{F}_{\text{SFE}} \in \mathbb{R}^{C_{\text{SFE}} \times H \times W \times D}$, where $C_{\text{SFE}}$ is the feature dimension. The feature map of each stage's SFE module is spatially center-cropped and passed as input to the next, producing more complex features in subsequent stages.

**Token initialization**. During token initialization, shallow features are projected and tokenized using differently-sized 3D image patches. Stages $L_1$, $L_2$ and $L_3$ use progressively larger patch sizes, covering broader context regions using the same number of tokens or less. Specifically, image token embeddings (ITEs) in each stage are produced by applying a convolution with a stride and kernel size of $p_i \times p_i \times p_i$, where $i \in \{1, 2, 3\}$ is the stage level. These tokens are then projected into vector embeddings of length $C_{\text{emb}}$ before being partitioned into 3D local attention windows of $M \times M \times M$ tokens. MTVNet employs specialized carrier tokens (CATs) which summarize the features within each local attention window. In each stage, carrier tokens are generated by applying a convolution with a stride and kernel size of $\lfloor \frac{M}{N_{cat}} \rfloor \times \lfloor \frac{M}{N_{cat}} \rfloor \times \lfloor \frac{M}{N_{cat}} \rfloor$ to the image tokens, yielding $N_{cat}^3$ carrier tokens of embedding length $C_{\text{emb}}$ for each attention window.

**Deep Feature Extraction**. For deep feature extraction in each stage, MTVNet employs groups of dense-connected hierarchical attention (DCHAT) blocks. Each group consists of $n \in \{1, 2, 3\}$ DCHAT blocks connected in a residual scheme, as shown in fig. 2. Cross-attention [58] is applied to merge the image tokens and carrier tokens of each DCHAT group into the subsequent stage's DCHAT group, facilitating propagation of multi-scale information.

**Reconstruction**. In the final stage, token upsampling is performed using deconvolution, transforming token embeddings back into the image space. These features are then refined in a pre-reconstruction stage before being fused with the shallow features from stage $L_1$ through a long skip-connection. The fused features are then upsampled using a 3D pixel-shuffle layer [59]. We employ a 3D pre-convolution layer initialized according to the ICNR method described in [60] to prevent checkerboard artifacts.

## 3.2 DCHAT block

For efficient extraction of volumetric image features, we propose a DCHAT block, see fig. 2. Inspired by DRCT [4], our DCHAT block employs a densely connected structure of volumetric transformer layers, LeakyReLU activations, and convolutions. To preserve the feature space of image tokens and carrier tokens, we process each token set using separate skip connections and convolutions. Additionally, we match the embedding dimension of all token embeddings throughout each block to equally promote learning of progressively complex features. As in DRCT [4], we utilize $1 \times 1 \times 1$ convolutions as gating mechanisms between transformer layers to filter redundant features, enabling direct feature transition between DCHAT blocks.

## 3.3 SVHAT layer

Inspired by FasterViT [21], we implement SVHAT (shifting volumetric hierarchical attention transformer) layer. Like FasterViT, SVHAT adopts the same use of specialized carrier tokens, which serve to summarize and propagate information between local attention windows. First, full attention of all carrier tokens enables global information exchange between attention window summaries. Then, each set of local window tokens is concatenated with their carrier tokens, and windowed attention is applied jointly, allowing carrier tokens to convey information from other windows. This alternating attention procedure efficiently transfers global information between local attention windows to improve information flow, see fig. A.2 in appendix which illustrates the intuition behind carrier tokens. To further enhance this, we reintroduce the notion of shifted-window attention from [6], see fig. 3. Before window partitioning, 3D cyclic-shifting is performed to allow the attention

of tokens in neighboring windows. To account for the presence of carrier tokens, we shift image tokens and carrier tokens by $\lfloor \frac{M}{2} \rfloor$ and $\lfloor \frac{N_{cat}}{2} \rfloor$ voxels, respectively, conserving the alignment of the token spaces. Attention masking is applied to drop interactions between non-adjacent image/carrier tokens.

We compute attended carrier token embeddings $\mathbf{x}_{\mathrm{cat}}^{L,t}$ at network level $L$ and transformer layer $t$ as:

$$\hat{\mathbf{x}}_{\mathrm{cat}}^{L,t} = \mathbf{x}_{\mathrm{cat}}^{L,t-1} + \gamma_1 \, \mathrm{MSA} \left( \mathrm{LN} \left( \mathbf{x}_{\mathrm{cat}}^{L,t-1} \right) \right),$$
$$\mathbf{x}_{\mathrm{cat}}^{L,t} = \hat{\mathbf{x}}_{\mathrm{cat}}^{L,t} + \gamma_2 \, \mathrm{MLP} \left( \mathrm{LN} \left( \hat{\mathbf{x}}_{\mathrm{cat}}^{L,t} \right) \right), \quad (1)$$

where $\gamma_1, \gamma_2$ are learnable channel-wise scaling factors, MSA denotes multi-headed self-attention [58], LN is Layer Normalization [61], and MLP is the multi-layer perceptron.

Next, we compute the attention of image tokens and carrier tokens using windowed self-attention, see eq. (2). Carrier tokens are window partitioned and concatenated with their corresponding set of local window tokens to produce sequences of $M^3 + N_{\mathrm{cat}}{}^3$ tokens for each window. Window-attended tokens $\mathbf{x}_{\mathbf{w}}^{L,t+1}$ are computed using post-normed shifted window self-attention (SW-MSA) [53] as:

$$\mathbf{x}_{\mathbf{w}}^{L,t} = [\mathbf{x}^{L,t-1}, \ \mathbf{x}_{\mathrm{cat}}^{L,t}]$$
$$\hat{\mathbf{x}}_{\mathbf{w}}^{L,t+1} = \mathbf{x}_{\mathbf{w}}^{L,t} + \mathrm{LN} \left( \mathrm{SW\text{-}MSA} \left( \mathbf{x}_{\mathbf{w}}^{L,t} \right) \right) \quad (2)$$
$$\mathbf{x}_{\mathbf{w}}^{L,t+1} = \hat{\mathbf{x}}_{\mathbf{w}}^{L,t+1} + \mathrm{LN} \left( \mathrm{MLP} \left( \hat{\mathbf{x}}_{\mathbf{w}}^{L,t+1} \right) \right)$$

The carrier tokens and image tokens are then separated for compatibility with later SVHAT layers.

Prior to the attention mechanisms described in eq. (1) and eq. (2), SVHAT uses multi-head cross-attention (MCA) layers to facilitate information exchange across network stages $L_1, L_2, L_3$. Each cross-attention layer implements a two-layer MLP to ensure dimension compatibility between cross-scale token sequences. Then, MCA is applied to capture relationships between tokens from current and previous network stages. Exploiting the compactness of the carrier token space, we compute cross-attended carrier tokens $\mathbf{x}_{\mathrm{cross, \, cat}}^{L}$ using full MCA:

$$\mathbf{x}_{\mathrm{cross, \, cat}}^{L} = \mathrm{LN} \left( \mathrm{MCA} \left( \mathbf{x}_{\mathrm{cat}}^{L,t-1}, \mathrm{MLP} \left( \mathbf{x}_{\mathrm{cat}}^{L-1} \right) \right) \right), \quad (3)$$

where $\mathbf{x}_{\mathrm{cat}}^{L-1}$ denotes the final set of carrier tokens from the previous network stage. A similar window-based multi-head cross-attention (W-MCA) mechanism is used for capturing relationships between image tokens, see equation 4. The cross-attended image tokens $\mathbf{x}_{\mathrm{cross}}^{L}$ are computed as:

$$\mathbf{x}_{\mathrm{cross}}^{L} = \mathrm{LN} \left( \mathrm{W\text{-}MCA} \left( \mathbf{x}^{L,t-1}, \mathrm{MLP} \left( \mathbf{x}^{L-1} \right) \right) \right), \quad (4)$$

where $\mathbf{x}^{L-1}$ denote the final set of image tokens from the previous network stage. Finally, the cross-attended token embeddings are fused by addition:

$$\mathbf{x}_{\mathrm{cat}}^{L,t-1} = \bar{\mathbf{x}}_{\mathrm{cat}}^{L,t-1} + \mathbf{x}_{\mathrm{cross, \, cat}}^{L}$$
$$\mathbf{x}^{L,t-1} = \bar{\mathbf{x}}^{L,t-1} + \mathbf{x}_{\mathrm{cross}}^{L} \quad (5)$$

Here, $\bar{\mathbf{x}}^{L,t-1}$ and $\bar{\mathbf{x}}_{\mathrm{cat}}^{L,t-1}$ denote image- and carrier tokens before fusion. For more details on the functionality of SVHAT, see appendix A.

# 4 Experiments

**Datasets**. We use four public MRI datasets and one CT-based dataset to train/evaluate our proposed MTVNet: The Human Connectome Project (HCP) 1200 Subjects dataset [62], the IXI dataset[1], the Brain Tumor Segmentation Challenge (BraTS) 2023 [63–66] and Kirby 21 [67]. All scans were acquired using 1.5T-3T MRI platforms with a volume size of $\leq 320^3$ voxels. Finally, we use the Femur Archaeological CT Superresolution (FACTS) dataset [68], which includes 12 registered 3D volume pairs of archaeological femur bones scanned using clinical-CT and micro-CT. The FACTS dataset consists of large volumes ($\sim 2000^3$ voxels) featuring detailed trabecular bone structures. Two SR tasks are considered using this dataset: In FACTS-Synth, we use downsampled micro-CT images as the LR model input, while FACTS-Real instead uses the clinical-CT images. Refer to appendix B for additional details.

**Models**. We evaluate the SR performance of MTVNet against 2D models RCAN [26] and HAT [2, 37], as well as six volumetric models: mDC-SRN [15], EDDSR [44], MFER [46], RRDBNet3D [27], SuperFormer [16], and ArSSR [45]. We adapt mDCSRN and SuperFormer, originally designed to restore images degraded by 3D k-space truncation [13, 16], by extending them with the upsampling module from MTVNet. We use the authors' suggested upsampling for the remaining models.

**Training**. We train all models from scratch on each dataset for 100K iterations on a single A100 80GB GPU. For ArSSR, we collate $N = 8000$ randomly sampled HR/LR point pairs from 15 patches per batch. The rest use batch size 4 for MRI or 5 for CT data. The LR patch size is set to $32 \times 32 \times 32$ or $32 \times 32$ in case of 2D. MTVNet $L_2$ and $L_3$ with two and three stages use patch sizes $64 \times 64 \times 64$ and $128 \times 128 \times 128$, respectively. All models are optimized using ADAM [69] with $\beta_1 = 0.9$ and $\beta_2 = 0.999$. We use a multi-step learning rate scheduler, halving the learning rate once after 50k, 70k, 85k, and 95k iterations. All model parameters are optimized using pure L1 loss. HR/LR pairs are generated using volumetric blurring followed by downsampling via linear interpolation. In FACTS-Real, we use downsampled clinical-CT images as the LR input and the micro-CT images as the HR reference.

**Evaluation**. We reconstruct all test samples from each dataset using strided aggregation of SR patch predictions. Patch predictions are tiled using an overlap of $4 \times s$ voxels where $s$ is the upscaling fac-

---

[1] https://brain-development.org/ixi-dataset/

| | FACTS-Synth Dataset | | | | | | FACTS-Real Dataset | | | | | |
| | Scale 4× | | | Scale 3× | | | Scale 4× | | | Scale 3× | | |
| 2D methods | PSNR | SSIM | NRMSE | PSNR | SSIM | NRMSE | PSNR | SSIM | NRMSE | PSNR | SSIM | NRMSE |
|---|---|---|---|---|---|---|---|---|---|---|---|---|
| RCAN [26] | 27.88 | .8940 | .1952 | 31.03 | .9336 | .1359 | 20.51 | .3554 | .5391 | 20.72 | .3870 | .5548 |
| † HAT [37] | 28.05 | .8951 | .1924 | 31.15 | .9334 | .1355 | 20.54 | .3686 | .5343 | 20.63 | .4242 | .5614 |
| **3D methods** | PSNR | SSIM | NRMSE | PSNR | SSIM | NRMSE | PSNR | SSIM | NRMSE | PSNR | SSIM | NRMSE |
| ArSSR [45] | 28.83 | .8998 | .1779 | 30.78 | .9284 | .1459 | 20.88 | .3871 | .4881 | 20.68 | .3980 | .5767 |
| EDDSR [44] | 29.86 | .9109 | .1620 | 33.22 | .9451 | .1104 | 20.62 | .3531 | .4815 | 19.84 | .3499 | .5223 |
| MFER [46] | 29.48 | .9094 | .1646 | 32.50 | .9420 | .1179 | 21.58 | .4708 | .4080 | 21.64 | .4671 | .4096 |
| mDCSRN [15] | 29.77 | .9099 | .1624 | 33.23 | .9460 | .1090 | 21.31 | .4078 | .4765 | 21.37 | .4259 | .4922 |
| † SuperFormer [16] | 30.46 | .9175 | .1481 | 33.47 | .9480 | .1055 | 20.93 | .3491 | .4846 | 21.40 | .4038 | .4463 |
| RRDBNet3D [27] | 29.78 | .9120 | .1584 | 33.21 | .9442 | .1093 | 21.64 | .4670 | .4022 | 21.91 | .4775 | .4019 |
| † MTVNet | 31.57 | .9303 | .1313 | 33.91 | .9502 | .1020 | 21.52 | .4576 | .4061 | 21.74 | .4633 | .4051 |

| | HCP 1200 Dataset | | | IXI Dataset | | | BraTS 2023 Dataset | | | Kirby 21 Dataset | | |
| | Scale 4× | | | Scale 4× | | | Scale 4× | | | Scale 4× | | |
| 2D methods | PSNR | SSIM | NRMSE | PSNR | SSIM | NRMSE | PSNR | SSIM | NRMSE | PSNR | SSIM | NRMSE |
|---|---|---|---|---|---|---|---|---|---|---|---|---|
| RCAN [26] | 32.61 | .8812 | .1593 | 28.52 | .8367 | .1768 | 33.47 | .9306 | .1505 | 33.06 | .8978 | .2256 |
| † HAT [37] | 32.39 | .8770 | .1640 | 28.42 | .8331 | .1791 | 33.20 | .9266 | .1551 | 31.18 | .8561 | .2940 |
| **3D methods** | PSNR | SSIM | NRMSE | PSNR | SSIM | NRMSE | PSNR | SSIM | NRMSE | PSNR | SSIM | NRMSE |
| ArSSR [45] | 27.90 | .8118 | .2810 | 24.22 | .7204 | .3060 | 22.96 | .3182 | .4437 | 31.82 | .8632 | .2775 |
| EDDSR [44] | 30.12 | .8335 | .2174 | 25.22 | .7394 | .2597 | 32.66 | .9169 | .1686 | 33.51 | .8946 | .2244 |
| MFER [46] | 33.40 | .8933 | .1484 | 25.23 | .7611 | .2576 | 34.76 | .9430 | .1309 | 35.68 | .9307 | .1719 |
| mDCSRN [15] | 33.46 | .8941 | .1470 | 29.50 | .8558 | .1622 | 34.76 | .9431 | .1308 | 35.26 | .9255 | .1806 |
| † SuperFormer [16] | 33.70 | .8982 | .1430 | 29.89 | .8679 | .1545 | 34.60 | .9400 | .1333 | 35.85 | .9341 | .1675 |
| RRDBNet3D [27] | 34.31 | .9092 | .1331 | 30.27 | .8793 | .1488 | 35.20 | .9486 | .1242 | 36.27 | .9376 | .1598 |
| † MTVNet | 34.04 | .9046 | .1374 | 30.16 | .8754 | .1502 | 35.16 | .9477 | .1250 | 35.97 | .9355 | .1654 |

**Table 1.** Quantitative comparison of state-of-the-art 2D/volumetric SR models on datasets FACTS-Synth, FACTS-Real, HCP 1200, IXI, BraTS 2023, and Kirby 21. The best performance metrics PSNR ↑ / SSIM ↑ / NRMSE ↓ are highligthed in red, and second best in blue. Transformer-based methods are marked with a † symbol.

tor, then smoothed using a Hanning window. Performance metrics Peak-Signal-to-Noise Ratio (PSNR), Structural Similarity Index Measure (SSIM), and Normalized Root Mean Square Error (NRMSE) are computed slice-wise in the axial direction and averaged over all samples, excluding slices where the foreground area occupies less than 20%.

## 4.1 Implementation details

All MTVNet configurations use a learning rate of $2e-4$ without weight decay. For MRI data, we use MTVNet $L_2$, as we found two stages to be enough to cover most whole scans. In the high-resolution FACTS dataset, we use MTVNet $L_3$ with three network stages. Each DCHAT block has 6 SVHAT layers. The number of shallow features $C_{SFE}$ and embedding features $C_{emb}$ are set to 128, with skip-connection features $C_{skip} = 64$. In MTVNet $L_3$, patch sizes are $p_1 = 2$, $p_2 = 4$, $p_3 = 8$; in $L_2$, $p_1 = 2$, $p_2 = 4$. The attention window size is $M = 8$, with $N_{cat} = 4$ carrier tokens. To reduce memory, we halve feature channels in MTVNet, mDCSRN, SuperFormer, and RRDBNet3D before upsampling.

## 4.2 Quantitative results

Table 1 compares MTVNet with eight SOTA SR models: RCAN, HAT, ArSSR, EDDSR, MFER, mD-CSRN, SuperFormer, and RRDBNet3D. Across all brain MRI datasets (HCP 1200, IXI, BraTS 2023, and Kirby 21), MTVNet achieves competitive results. We observe the CNN-based RRDBNet3D slightly outperforming the ViT-based MTVNet and Super-Former on brain MRI, while in 2D, RCAN similarly surpasses the newer ViT-based HAT. This trend suggests that for low-resolution data, CNNs outperform ViTs, contradicting earlier findings [16]. We reason that the advantage of CNNs in these datasets stems from a combination of low image resolution and local image dependencies being predominant, limiting the benefits of the broader receptive field offered by ViTs.

In the high-resolution FACTS dataset, where we can leverage the multi-contextual architecture of our proposed method, we observe several new trends: in FACTS-Synth, ViT-based methods surpass CNN-based architectures in both 2D and 3D, with MTVNet outperforming all methods by a large margin. Compared with SuperFormer, MTVNet improves PSNR by 0.44dB–1.11dB, and by 0.70dB–1.79dB over RRDBNet3D, illustrating that added contextual information yields significant gains in high-resolution volumetric SR. In FACTS-Real, where clinical CT images serve as LR input, the best results are achieved by RRDBNet3D, MFER and MTVNet, despite the similarity to FACTS-Synth. We hypothesize this stems from the domain shift between micro-CT and clinical-CT, which weakens

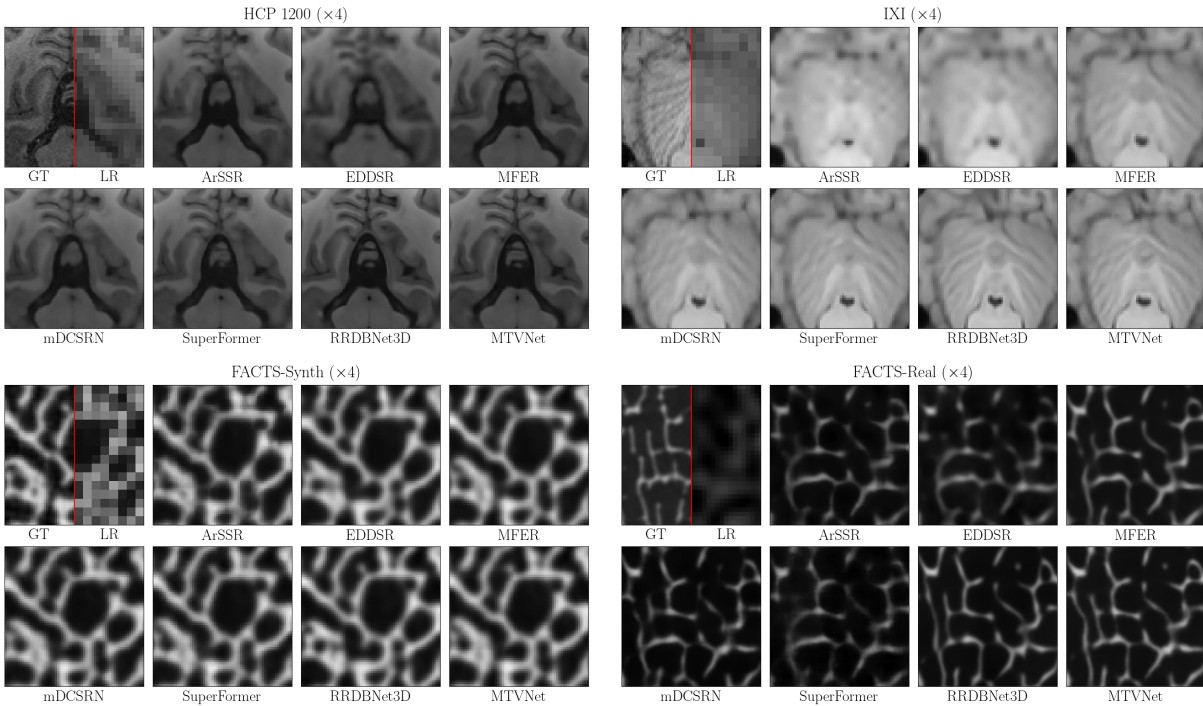

**Figure 4.** Visual comparisons of SR outputs on HCP 1200, IXI, FACTS-Synth, and FACTS-Real at 4× upscaling. Ground truth (GT) and LR inputs are shown in the top-left, separated by a red line.

long-range dependencies that would otherwise benefit ViTs. Overall, these results indicate that CNN vs. ViT performance depends strongly on both input resolution and on the underlying 3D image structure.

## 4.3 Qualitative results and 3D LAM

Fig. 4 shows a visual comparison of SR predictions on scale 4× for HCP 1200, IXI, FACTS-Synth, and FACTS-Real. We find that MTVNet produces faithful reconstructions of structures and patterns across all datasets. Compared with ArSSR, EDDSR, MFER, mDCSRN, and SuperFormer, our MTVNet produces notably sharper features while producing similar results as RRDBNet3D. In the Brain MRI datasets HCP 1200 and IXI, we find that many methods struggle to reconstruct anatomical details while RRDBNet3D and our MTVNet produce the clearest results. Refer to appendix E for more comparisons.

Next, we investigate how volumetric SR models leverage surrounding image context using LAM [70]. We extend LAM into 3D to visualize context usage in volumetric SR predictions. Fig. 5 shows log-scaled LAM activations on FACTS-Synth at ×4 upscaling, where higher intensities indicate stronger voxel contributions towards the region marked by the red box. The blue box highlights the SR prediction area, which is constant across methods. Although no predictions are computed outside this area, MTVNet can use information from these regions via its contextual stages. To quantify context usage, we compute the Diffusion Index (DI) [70]. We report the mean DI across slices for volumetric SR methods and per-slice DI for 2D methods. Examples from FACTS-Synth show that MTVNet, with contextual stages, enables broader leverage of input context than competing methods. Additional LAM comparisons are provided in appendix C. To study the importance of context across datasets, we compute average DI scores over 50 random 3D patch samples from HCP 1200 and FACTS-Synth, see fig. 6. DI scores are generally lower in HCP 1200, especially for stronger models, suggesting that context is less critical in brain MRI. Conversely, SR models achieve higher DI in FACTS-Synth, with MTVNet achieving the highest average DI among all methods.

## 4.4 Ablation experiments

We perform ablation of the features of MTVNet, including carrier tokens and contextual network stages across MRI and CT data. Table 2 shows a quantitative comparison on BraTS 2023 and FACTS-Synth using ×4 upscaling. Using BraTS 2023, replacing the baseline Swin transformer layers [6] with our SVHAT layers using carrier tokens results in modest performance gains across all metrics. Using FACTS-Synth, increasing the number of context levels of MTVNet results in significant performance improvements across all metrics. Compared with MTVNet $L_1$, adding an extra level of context increases PSNR by 0.44dB. Similarly, using three contextual network stages further improves PSNR by 1.1dB.

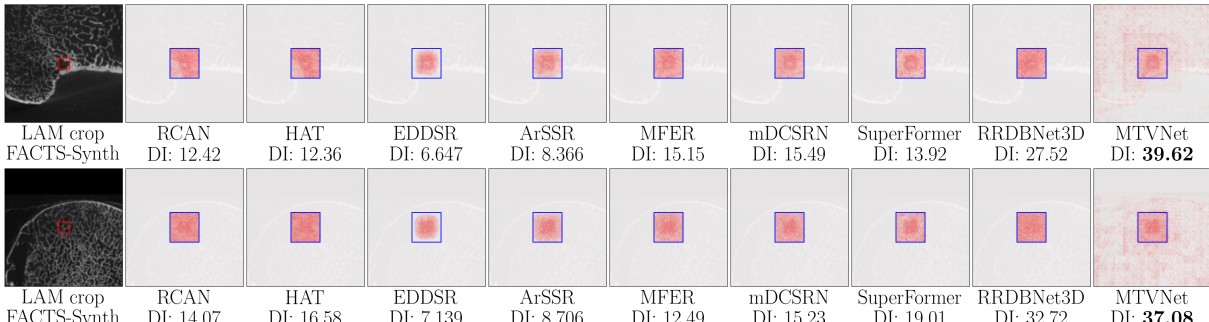

**Figure 5.** LAM comparisons of SR models using FACTS-Synth at ×4 upscaling. The blue box marks the prediction area for SR, which is the same for all methods. The highest DI ↑ is highlighted in **bold**.

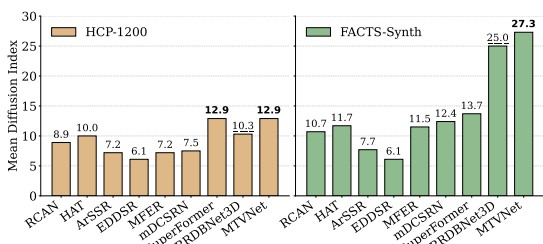

**Figure 6.** DI score averages across 50 random 3D samples from HCP 1200 (×4) and FACTS-Synth (×4).

| Method | #Params | CATs | PSNR/SSIM/NRMSE |
|---|---|---|---|
| Baseline | 19.7M | ✗ | 35.00 / .9460 / .1273 |
| MTVNet $L_1$ | 34.6M | ✓ | 35.05 / .9467 / .1265 |

| Method | #Params | Levels | PSNR/SSIM/NRMSE |
|---|---|---|---|
| MTVNet $L_1$ | 34.6M | 1 | 30.03 / .9168 / .1550 |
| MTVNet $L_2$ | 109.0M | 2 | 30.47 / .9211 / .1452 |
| MTVNet $L_3$ | 138.2M | 3 | 31.57 / .9303 / .1313 |

**Table 2.** Ablation on the effect of carrier tokens and context levels. Best metrics are underlined.

## 4.5 Memory footprint of MTVNet

Fig. 7 shows the memory footprint of SuperFormer, RRDBNet3D, and MTVNet across volumetric input resolutions. Memory footprint is measured as the peak GPU memory usage for one forward and backward pass using a batch size of 1. With one network stage, MTVNet $L_1$ exhibits better memory scaling than SuperFormer and RRDBNet3D. Provided the prediction area is fixed to $32^3$, adding contextual network stages allows processing of input sizes far exceeding the capabilities of other architectures.

Fig. 8 shows PSNR vs. throughput of Super-Former, RRDBNet3D, and MTVNet on FACTS-Synth using 4× upscaling. MTVNet achieves SOTA performance while maintaining a higher throughput than SuperFormer, despite having more parameters.

## 5 Conclusion

In this work, we present MTVNet, a ViT-based method for volumetric SR tailored for high-

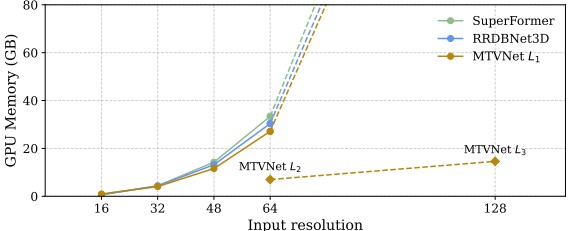

**Figure 7.** GPU memory usage across 3D patch resolutions. Contextual stages in MTVNet enable resolutions of $128^3$ and higher without exceeding VRAM limits.

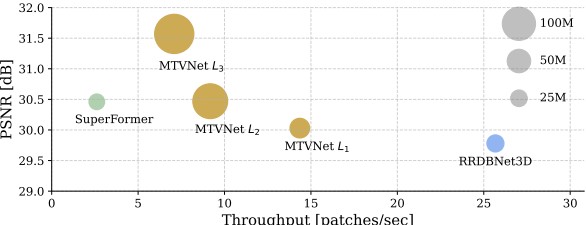

**Figure 8.** PSNR vs. Throughput using FACTS-Synth (4×). Throughput is measured on a NVIDIA H100 GPU using a batch size of 1.

resolution 3D data. Our method overcomes the challenge of limited contextual information through a multi-contextual network structure with a coarse-to-fine feature extraction and tokenization scheme, enabling processing of larger input sizes than competing methods. We model long-range dependencies by combining global and window-based attention to exchange information in a larger input volume.

We compare MTVNet against 2D and volumetric SR approaches across several data domains, including brain MRI data and high-resolution CT data. Based on extensive experiments, we find that CNN-based models outperform ViT-based models in certain 3D data domains. CNN-based SR models are especially effective in low-resolution 3D volumes where the receptive field of transformers cannot be leveraged as effectively. Nevertheless, our proposed MTVNet with extra contextual processing layers outperforms all other models in high-resolution 3D data with long-range image dependencies.

# Acknowledgments

Research reported in this publication is supported by the Infrastructure for Quantitative AI-based Tomography (QUAITOM) supported by the Novo Nordisk Foundation (Grant number NNF21OC0069766) and the Multiscale label-free 3D x-ray imaging: Visualizing cells and tissue architecture simultaneously (Xtreme-CT) supported by the Novo Nordisk Foundation (Grant number grant NNF22OC0077698).

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

# A  Details of SVHAT layer

An overview of our proposed SVHAT layer featuring separate attention branches for carrier tokens and image tokens is illustrated in fig. A.1. The first branch (red) follows the attention procedure from FasterViT [21], whereas the second branch follows the procedure from SwinV2 [53] with post-normalization. We use multi-head cross attention (MCA) and window-based multi-head cross attention (W-MCA) to merge tokens from previous network stages before computing attention in each branch. Embedding dimensions from previous network stages are matched using a small MLP.

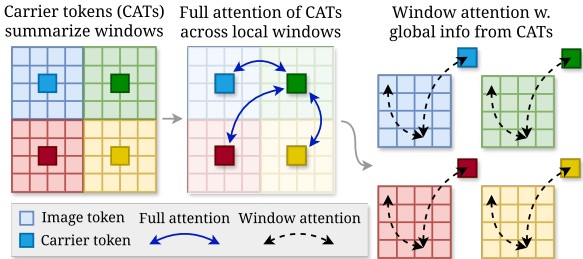

**Figure A.2.** Illustration of the carrier token (CAT) attention mechanism proposed by Hatamizadeh et al. [21] and adopted by SVHAT. Carrier tokens convey global information between local attention windows to enrich long-range information propagation.

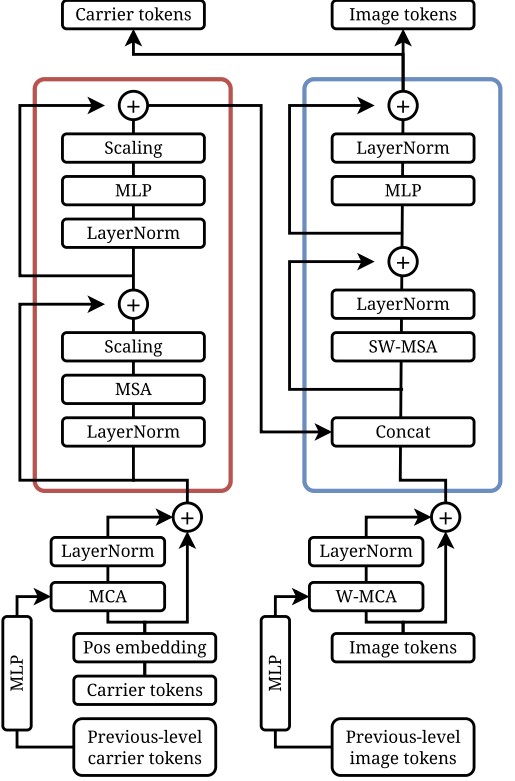

**Figure A.1.** Overview of our proposed SVHAT layer that captures global and local token dependencies using separate attention branches for carrier tokens (red) and image tokens (blue).

An illustration of the functionality of carrier tokens is provided in fig. A.2, showing both full CAT attention and local window attention with CATs.

# B  Dataset details

**Human Connectome Project**
The Human Connectome Project (HCP) 1200 Subjects Data Release [62] includes structural MRI scans from 1113 healthy subjects acquired using a 3T scanning platform. We use the T1-weighted images, featuring an isotropic resolution of 0.7 mm and a matrix size of $320 \times 320 \times 256$. Following [15, 16],

the dataset is split into 780 subjects for training, and 111 each for validation, evaluation, and testing. Performance evaluation is conducted on the test set.

**Information eXtraction from Images**
The Information eXtraction from Images (IXI) dataset contains multi-modality MRI data (PD-, T1-, and T2-weighted) from 600 healthy subjects scanned with one 3T and two 1.5T scanning platforms. We use 581 T1-weighted scans, of which 507 have a resolution of $0.9375 \times 0.9375 \times 1.2$ mm and a matrix size of $256 \times 256 \times 150$, while the remaining 74 have a similar resolution but a matrix size of $256 \times 256 \times 146$. The dataset is split into 500 subjects for training, 6 for validation, and 75 for testing, with evaluation performed on the test set.

**Brain Tumor Segmentation Challenge 2023**
For the Brain Tumor Segmentation Challenge (BraTS) 2023, we use 1470 T1-weighted skull-stripped MRI scans of glioma patients, standardized to an isotropic resolution of 1 mm and a matrix size of $240 \times 240 \times 155$. We use the dataset split provided by the challenge, which allocates 1251 subjects for training and 219 for validation, with evaluation performed on the validation set.

**Kirby 21**
The Kirby 21 dataset includes multi-modality MRI scans from healthy individuals with no history of neurological conditions. We use the 42 T2-weighted images, which have a resolution of $1 \times 0.9375 \times 0.9375$ mm and a matrix size of $180 \times 256 \times 256$. The data is split into 37 images for training (KKI-06 to KKI-42) and 5 for testing (KKI-01 to KKI-05).

**Femur Archaeological CT Superresolution**
The Femur Archaeological CT Superresolution (FACTS) dataset comprises 12 archaeological proximal femurs scanned with clinical-CT and micro-CT platforms [68]. Clinical-CT scans have a resolution of $0.21 \times 0.21 \times 0.4$ mm, while micro-CT scans have a resolution of $58 \times 58 \times 58$ $\mu$m. Clinical-CT volumes are registered and linearly interpolated to match the micro-CT matrix size. The dataset is split into 10 images for training and 2 (f_002 and f_138) for testing.

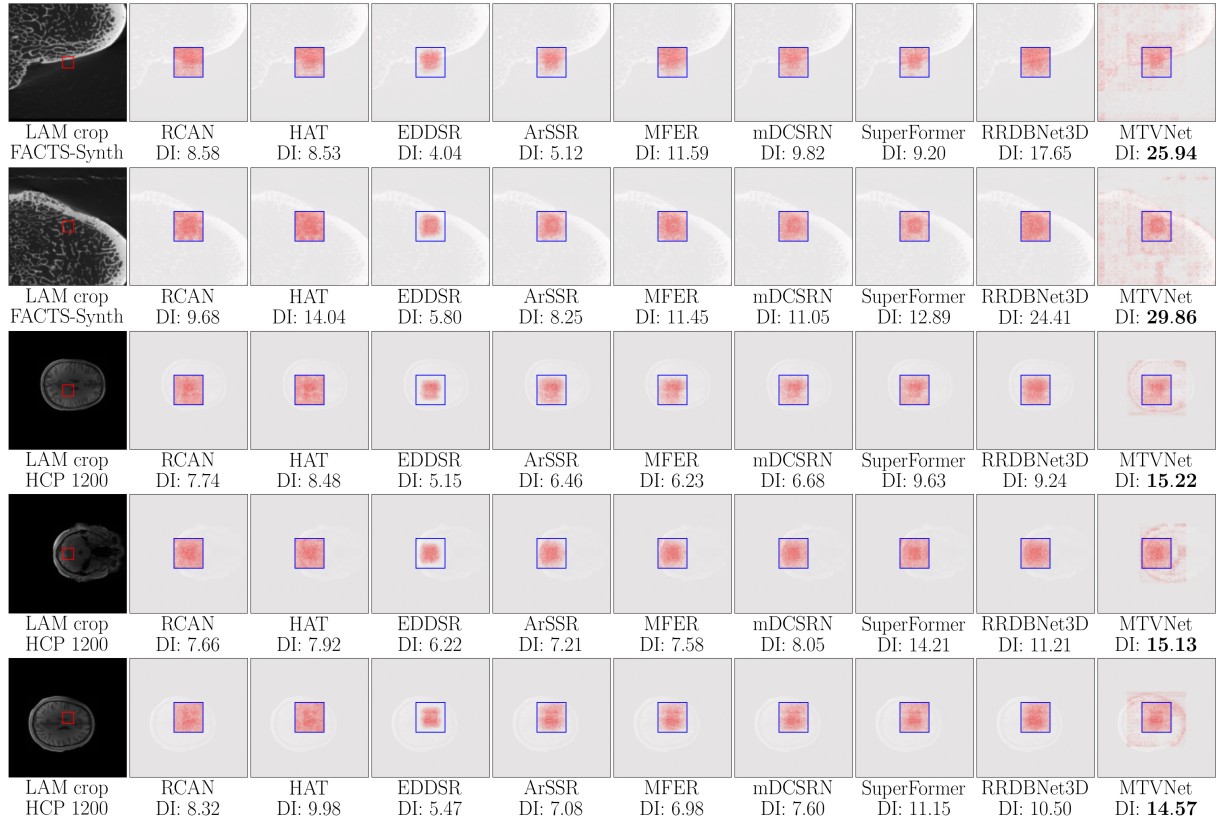

**Figure B.1.** LAM comparisons of SR models using FACTS-Synth and HCP 1200 at ×4 upscaling. The blue box shows the SR prediction area, which is kept constant across all methods. The highest DI ↑ is highlighted in **bold**.

## C  More LAM comparisons

Additional visual comparisons of the LAM method using FACTS-Synth and the HCP 1200 dataset at ×4 upscaling are shown in figure B.1. Similar to FACTS-Synth, we find that MTVNet incorporates information from the surrounding image context when trained using the HCP 1200 dataset. Despite this, we observe the CNN-based RRDBNet3D achieving better performance, suggesting that image context is less critical in brain MRI data.

## D  MTVNet hyperparameters

Tab. E.1 shows an overview of the GPU memory usage, throughput and parameter count of MTVNet using different hyperparameter configurations. Memory usage is measured as the maximum GPU memory required for a single forward and backward pass using a batch size of 1. The throughput in patches/sec is measured using a batch size of 1 assuming 4× upscaling. All configurations use MTVNet $L_3$ with three network stages. The number of blocks in table E.1 denotes the total number of DCHAT blocks used, with 6 blocks corresponding to a depth of $(1, 2, 3)$ for network stages $(L_1, L_2, L_3)$, 9 blocks corresponding to network depths $(2, 3, 4)$, and 12 blocks corresponding to network depths $(3, 4, 5)$.

| Parameter | Memory usage | Throughput | #Params |
|---|---|---|---|
| $C_{\text{skip}}$ | | | |
| 64* | 10.47 GB | 7.79 patches/sec | 138.3M |
| 96 | 10.83 GB | 7.70 patches/sec | 226.6M |
| 128 | 11.24 GB | 7.50 patches/sec | 340.3M |
| $N_{cat}$ | | | |
| 4* | 10.47 GB | 7.79 patches/sec | 138.3M |
| 2 | 10.31 GB | 7.75 patches/sec | 141.0M |
| 1 | 10.37 GB | 7.90 patches/sec | 163.0M |
| #Blocks | | | |
| 6* | 10.47 GB | 7.79 patches/sec | 138.3M |
| 9 | 10.98 GB | 5.51 patches/sec | 198.2M |
| 12 | 11.49 GB | 4.25 patches/sec | 258.1M |
| $C_{\text{emb}}$ | | | |
| 128* | 10.47 GB | 7.79 patches/sec | 138.3M |
| 192 | 10.03 GB | 7.47 patches/sec | 194.5M |
| 256 | 10.30 GB | 7.34 patches/sec | 261.3M |

**Table E.1.** Overview of memory usage, throughput and no. of parameters using different hyperparameter configurations of MTVNet $L_3$. Baseline parameters of MTVNet $L_3$ are highlighted with an asterisk.

## E  More visual comparisons

Additional visual comparisons of SR predictions for HCP 1200, IXI, BraTS 2023, Kirby 21, FACTS-Synth, and FACTS-Real are shown in fig. E.1.

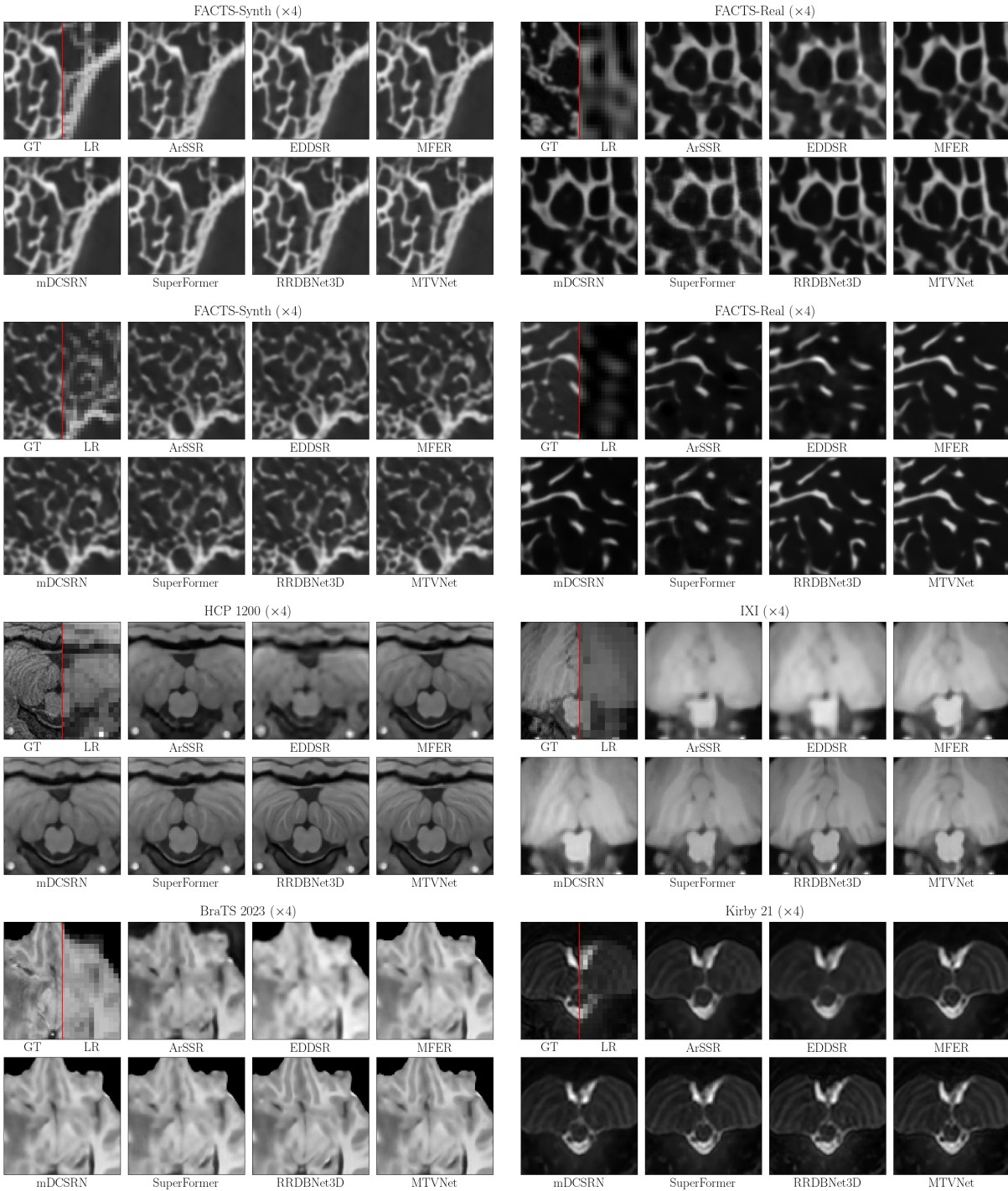

**Figure E.1.** Visual comparisons of SR outputs on HCP 1200, IXI, BraTS 2023, Kirby 21, FACTS-Synth, and FACTS-Real at 4× upscaling. Ground truth (GT) and LR inputs are shown in the top-left, separated by a red line.

