# OpenReview forum: "MTVNet: Multi-Contextual Transformers for Volumes – Network for Super-Resolution with Long-Range Interactions"
_NLDL.org/2026/Conference — NLDL 2026 Spotlight_

### Official Review · Reviewer_qmrh · 2025-09-16
**MTVNet Review NLDL 2026**

**Rating:** 4
**Confidence:** 3

**Summary:**

This paper presents MTVNet, a novel network structure to bring the success of super-resolution in 2D images to a 3D environment using ViTs as opposed to CNNs. This article wants to use ViTs over CNNs due to their receptive fields being greater, thus allowing for more contextual clues. While ViTs have surpassed CNN models in the 2D setting, there still exists a performance gap in the 3D/volumetric setting due to how self attention is affected by 3D datastructures.
The contributions of this article are as follows:
1. Present MTVNet as a solution to the problem of self attention in 3D data by utilizing hierarchical attention mechanisms and multi-contextual information
1.2. The contribution of MTVNet includes dense-connected hierarchical attention (DCHAT) and the Shifting-Volumetric hierarchical attention transformer (SVHAT)
2. Investigate the performance gap between CNNs and ViTs based methods for 3D data.

The MTCNet works in three stages which the authors denotes as $L_{1}$, $L_{2}$, $L_{3}$. These stages corresponds to different context levels, being fine, medium and coarse respectively. The most relevant thing to note with $L_{3}$ and $L_{2}$ is that they incorporate areas outside of the predicted/reconstructed area to provide more context for the model. This gives stage $L_{1}$ some priors to work with when producing the super-resolution output of a given area.
The features from the different stages are  sent through a shallow feature extractor, where parts of it is combined with the finer context level and the full output is sent through a DCHAT block.

This article uses four public MRI datasets in their experiments and a femur CT dataset. They compare MTVNet with two 2D super-resolution models, RCAN and HAT, as well as six other volumetric models. Model performance are measured by Peak-Signal-to-Noise ratio (PSNR), Structural Similarity Index Measure (SSIM) and Normalized Root Mean Square Error (NRMSE).

The results provided show MTVNet outperforming all compared methods on the CT synthetic dataset, while performing comparable to the best performers on the real CT data.
For the MRI datasets we see the MTVNet being the second best over all datasets, just behind the CNN based RRDBNet3D model. From the results the authors notes a trend where CNN based super-resolution architectures outperforms ViT based models on low-resolution images due to the limitations to the benefits of broader receptive fields offered by ViTs. The authors also showcase several examples from different datasets where their model has made what seems like a faithful reconstruction of the original image.

Then the authors explore the different models using a method called LAM which is based on the same concepts as integrated gradient and showcases pixels which has attributed to the recreation of a given area in the super-resolution image. This section shows the authors claim of their model using larger contexts to recreate a given area over the other, more limited models.

Lastly the authors compares the memory footprint of different models, showcasing the efficiency of their proposed MTVNet.

**Strengths:**

$\textbf{Methods}$
The method section is clear and explains the important components to a satisfactory level. I find the explanations of the shallow feature extractor, token initialization and DCHAT block to be easy enough to understand. I had some problems with the SVHAT layer explanation, but I am willing to say it is likely me who struggle to wrap my head around all the moving parts.

$\textbf{Experiments}$
I find the experiment section of this article to be its strong point. The experimental setup is clear, and I get a reasonable idea of the datasets in use. The results are communicated clearly, with both the top performing model and the second best performing model being easy to find for the different experimental setups.
The inclusion of the LAM attribution maps also helps strengthening the argument that your model does use context outside of the area of interest as opposed to the other models which uses more localized context.
Finally the memory footprint I am not sure how relevant it is to include, but I get this might be a background/field specific thing. This section does show that the proposed model is memory efficient, which is nice.

**Weaknesses:**

$\textbf{Introduction}$
My biggest problem and critique of this paper is the lack of motivation and story in the paper. To put it bluntly; I would like to see in the introduction justify the paper itself.
Do not get me wrong, I think this article presents work which is interesting. I have however no idea why medical images are being used, why I should care about super-resolution.
There is an attempt at motivating the article in paragraph 3 in the introduction, but it is severely lacking. The lack of motivation makes the article boring, which is a shame as I think the results are worthy of being shared.
I believe this article needs a small rewrite in the introduction with a focus on its motivation, and to make the story this article wants to tell more clear. I want to be exited to see how you want to bring ViTs to the level of CNNs in volumetric super-reolution, but I find it hard with this introduction. I also want to know why the focus on medical data. Why is super-resolution important for clinical use? Is it because of bad MRI machines? Are medical imaging often low-resolution? This is not obvious.
If this is a space issue then I would say put the memory section of the result in the appendix and focus on the story of the article.

$\textbf{Methods}$
No comments

$\textbf{Experiments}$
I am wondering why the $L_{3}$ module was only used on the CT data and not also on the MRI data. I might have missed the explanation somewhere, but would we not want to use more context than less? This is not a major point, just something I thought I'd point out.

I do also wonder why use medical data and not something more notorious for having large images to take advantage of the wide reception field super-resolution model. Would satellite imaging make more sense to showcase the strength of this model? I think this again comes back to a lack of proper motivation in the introduction, as to why this work is relevant for a medical setting.

**Justification:**

I think the article provides a novel model which is worth sharing. I think the experiments are solid, and interesting. The results, while not SotA, shows a clear improvement on other similar works and as such deserves that recognition.
I would have liked to see a lot more motivation into the paper itself. If I could give a rating of "need small changes" I absolutely would, as some of the motivations provided in the introduction seems incomplete.
If the motivation was more clear I would think this text was much easier to read. I will put my verdict as accept as I think the results are interesting, and hope my critique on the lacking motivation makes in into future works.

---

> ### Author Rebuttal · Authors · 2025-10-21
>
> We are very pleased with the reviewer's constructive input and useful suggestions for improving our work. We appreciate the reviewer finding most parts of our method section to be accessible and easy to understand. We are also glad that the reviewer finds our experimental section to be a strong point of the paper, with clearly communicated results and meaningful comparisons between models. We also thank the reviewer for highlighting the usefulness of the LAM section of our paper for illustrating how MTVNet utilizes contextual information beyond the prediction area, unlike competing methods.
>
> Motivation in relation to medical imaging: We appreciate the reviewer’s request for stronger motivation regarding the use of super-resolution in medical imaging. We believe this application is relevant, as the majority of 3D datasets come from medical modalities such as MRI and CT [1]. Volumetric super-resolution has the potential to improving diagnostic accuracy and treatment planning, enabling reduced patient radiation dose, and lowering hospital costs by extending the utility of existing scanners. These factors make medical imaging a natural and impactful domain for advancing volumetric super-resolution. We will improve the introduction to put more emphasis on motivation in the final manuscript.
>
> Use of $L_{3}$ module in MRI data: We understand the reviewers comments regarding the use of the $L_{3}$ module. We reasoned that employing the $L_{3}$ module on MRI data would incur unnecessary computational costs, as the $L_{2}$ module already extends beyond the spatial boundaries of most MRI scans due to their low resolution. Therefore, we limited the use of $L_{3}$ to higher resolution CT data, where a larger receptive field would be more beneficial. We will clarify this design choice in the final version of the paper.
>
> Choice of image domain: We thank the reviewer for their recommendation regarding the applicability of our work. We agree that satellite imaging represents an interesting and promising application for ViT models featuring large receptive fields. Yet, satellite images represent a different domain from volumetric images, and have unique computational challenges. Satellite images often contain a high number of channels from multiple modalities, and the methods considered in this paper would need to be adjusted to accommodate this. We argue that our work is therefore not directly applicable in this domain. We believe that our proposed method could also find applicability in the field of synchrotron imaging, where high-resolution volumetric images are commonly encountered.
>
> [1] Z. Ji, et al. “Deep learning-based magnetic resonance image super-resolution: a survey”. In: Neural Computing and Applications (2024), pp. 1–28.

---

### Official Review · Reviewer_KGvZ · 2025-10-02
**A solid paper with a novel approach to volumetric super-resolution, but with some weaknesses in the experimental evaluation.**

**Rating:** 4
**Confidence:** 5

**Summary:**

This paper introduces MTVNet, a novel 3D transformer-based architecture for volumetric super-resolution (SR). The main contribution is a multi-scale, multi-context approach that aims to increase the receptive field of the model without incurring the high computational cost of standard 3D self-attention. The model processes a large contextual volume at multiple resolutions, with coarser resolutions providing context for the finer-resolution SR prediction in the center. The authors also introduce a hierarchical attention mechanism (SVHAT) inspired by FasterViT. The experiments on five 3D datasets show that MTVNet outperforms existing methods on high-resolution data, while CNNs perform better on low-resolution data.

**Strengths:**

1.  The paper is written in a clear and accessible manner, making it easy to understand. The architecture of MTVNet is clearly presented in Figure 2. Moreover, the SVHAT layer is vividly depicted in Figure 3.
2.  The work has the potential to be significant for the field of medical imaging and other areas where high-resolution volumetric data is common. The finding that the performance of ViTs versus CNNs is dependent on the data resolution is also an interesting and important contribution.
3.  The proposed method is technically sound. The design of the DCHAT block and the SVHAT layer seems reasonable.

**Weaknesses:**

1.  The paper would be stronger with more extensive ablation studies. For example, what is the contribution of each component of MTVNet (e.g., the multi-context approach, the DCHAT block, the SVHAT layer)?
2.  The authors repeatedly claim throughout the paper that, in contrast to other transformer - based approaches, their method can expand the receptive field (as indicated in Line 22, 449, and 561). To strengthen this claim, it is advisable to present either quantitative analysis or visualizations. Such quantitative analysis could involve specific metrics that measure the receptive field size, and visualizations could offer a more intuitive understanding of how the receptive field is expanded.
3.  Throughout the paper, it is repeatedly stated that SVHAT is inspired by FasterViT (as noted in Line 95 and 290). To more convincingly demonstrate the superiority of SVHAT over FasterViT, it is highly recommended to conduct ablation experiments. These experiments could isolate and analyze the key components of SVHAT and FasterViT, enabling a direct comparison of their performance.

**Justification:**

This is a good paper that proposes a novel and interesting approach to volumetric super-resolution. The method is well-motivated and the paper is clearly written. The experimental results are promising and suggest that the proposed method can achieve state-of-the-art performance. However, the paper could be improved by including more extensive ablation studies.

---

> ### Author Rebuttal · Authors · 2025-10-21
>
> We are grateful for the reviewer's remarks and insights regarding our work. We greatly appreciate the reviewer's opinion on the accessibility of our writing and the clarity of our illustrations of the overall architecture of MTVNet and proposed SVHAT layer. We are also pleased that the reviewer finds our proposed method to be technically sound and recognizes the potential impact of our work in medical imaging and other domains involving high-resolution volumetric data. Finally, we appreciate the reviewer highlighting the importance of our finding that the relative performance of ViTs and CNNs depends on data resolution.
>
> Ablation study: We appreciate the reviewer's comments on the ablation study of MTVNet and its components. Our current ablation study already isolates the impact of the multi-contextual network stages, and the addition of carrier tokens used by the DCHAT block and SVHAT layer. As noted by another reviewer, additional details regarding hyperparameter selection and their impact on memory usage could be useful to some readers. We will extend the appendix with additional details on this in the final manuscript.
>
> Receptive field quantification: We agree with the reviewer’s remark on the relevance of quantifying the expanded receptive field of our proposed method. To address this, we employ the LAM [2] method and visualize the contribution of the surrounding image context on super-resolution predictions, see Fig. 5. We use the Diffusion Index (DI) as a metric to quantify the overall involvement of image context. Our results show that MTVNet has the broadest overall receptive field compared with other methods attributed to its multi-contextual network structure.
>
> Relation to FasterViT: We understand the reviewer's interest in the relation between SVHAT and FasterViT [1]. While SVHAT leverages the carrier token concept introduced in FasterViT, the two transformer architectures serve fundamentally different purposes: FasterViT was developed for 2D image classification, whereas SVHAT is designed for volumetric super-resolution. Hence, a direct comparison would not be meaningful. Our contribution lies in adapting the carrier token idea to the domain of volumetric data. The SVHAT layer is therefore not a direct improvement of FasterViT’s HAT layer, but rather a novel adaptation that enables efficient 3D image processing. We will make this distinction clearer in the final manuscript.
>
> [1] Hatamizadeh, Ali, et al. "FasterViT: Fast vision transformers with hierarchical attention.", The Twelfth International Conference on Learning Representations (2024).
>
> [2] Gu, Jinjin, and Chao Dong. "Interpreting super-resolution networks with local attribution maps." Proceedings of the IEEE/CVF conference on computer vision and pattern recognition (2021).

---

### Official Review · Reviewer_ohW5 · 2025-10-02

**Rating:** 4
**Confidence:** 4

**Summary:**

This paper proposes MTVNet, a transformer-based architecture designed for volumetric super-resolution (SR). The core idea is to enable efficient long-range interaction in 3D data by leveraging multi-contextual coarse-to-fine modeling with carrier tokens. The authors demonstrate that their approach significantly enlarges the effective receptive field of ViT-based models while maintaining memory feasibility. Experiments across multiple datasets (MRI and CT, including FACTS-Synth and HCP 1200) show that MTVNet achieves state-of-the-art performance in high-resolution volumetric SR, outperforming existing CNN- and ViT-based baselines. The paper is well-motivated and addresses an important gap in extending 2D ViT advantages to volumetric data.

**Strengths:**

1. The paper proposes a coarse-to-fine design with multi-scale contextual modeling to address the cubic growth of token numbers in 3D ViTs.
2. The paper evaluates the method on MRI, CT, and multiple other datasets, covering both real and synthetic scenarios, demonstrating the generalizability of the approach and showing strong performance.

**Weaknesses:**

1. There is a trade-off between efficiency and performance in the proposed method; it is recommended to compare with more recent lightweight ViT or CNN-ViT approaches to better demonstrate the method’s justification.

2. On low-resolution 3D data, CNNs still outperform ViTs, indicating that the proposed method’s advantages are mainly in high-resolution scenarios, which limits its applicability.

3. The model is relatively large, with up to 138M parameters. Although memory scalability is improved, it may still be challenging to deploy in resource-constrained environments such as clinical settings.

4. It is recommended to provide more intuitive explanations or visualizations to clarify the functional mechanism of the carrier tokens.

**Justification:**

The paper presents a contribution to volumetric SR with transformers, offering a well-designed approach that achieves state-of-the-art results in high-resolution 3D imaging.

---

> ### Author Rebuttal · Authors · 2025-10-21
>
> We would like to thank the reviewer for their constructive feedback on our work. We appreciate the reviewer acknowledging the completeness of our experiments across both MRI and CT datasets. We are pleased that the reviewer recognizes the performance and generalizability of our proposed method, and for highlighting the strength of our work addressing the bottleneck of cubic growth of tokens in 3D ViTs.
>
> Comparison with other approaches: We recognize that including additional methods for comparison could further strengthen the paper. Currently, there is a lack of ViT-based methods designed for volumetric super-resolution [3]. While recent ViT-based axial super-resolution methods do exist [1], they are not directly comparable with our approach. In our experiments, we found 3D methods to be overall superior to 2D approaches, and therefore found it sufficient to limit the inclusion of 2D methods. We argue that our study already includes recent methods, including MFER (2024), HAT (2023) and SuperFormer (2022). Although RRDBNet3D is from 2019, it remains the strongest baseline across MRI benchmarks, justifying its inclusion.
>
> Performance of CNNs vs. ViTs: We agree with the reviewer's comment that our method’s advantages are most pronounced in high-resolution data, while CNNs remain stronger in low-resolution data. Still, as remarked by another reviewer, the finding that the performance of ViTs vs. CNNs depends on data resolution, is an important contribution.
>
> Clinical applicability: We acknowledge the reviewer’s concern regarding the model’s size and its potential deployment challenges in clinical environments. While our method contains a relatively large number of parameters, its memory scalability and reduced token complexity make it less sensitive to input resolution than conventional 3D ViTs.
>
> Explanation of carrier tokens: We appreciate the suggestion to provide more intuitive explanations of the carrier token mechanism. We recognize that the use of carrier tokens may be difficult to grasp, even for readers familiar with ViT-based architectures. In the final manuscript, we will include a more intuitive explanation of how carrier tokens facilitate global information propagation. The use of carrier tokens is inspired by FasterViT [2], and we encourage interested readers to consult that work for further technical details.
>
> [1] Huang, Shan, et al. "TransMRSR: transformer-based self-distilled generative prior for brain MRI super-resolution." The Visual Computer 39.8 (2023): 3647-3659.
>
> [2] Hatamizadeh, Ali, et al. "FasterViT: Fast vision transformers with hierarchical attention.", The Twelfth International Conference on Learning Representations (2024).
>
> [3] Z. Ji, et al. “Deep learning-based magnetic resonance image super-resolution: a survey”. In: Neural Computing and Applications (2024), pp. 1–28.

---

### Official Review · Reviewer_v6ki · 2025-10-03
**The proposed method is scalable and effective but needs further clarifications on technical details**

**Rating:** 4
**Confidence:** 3
**Final Rating:** 4
**Final Confidence:** 4

**Summary:**

This paper presents a transformer-based model for volumetric super-resolution tasks, called MTVNet. This method aims to mitigate the computational memory issue for 3D data. The authors propose a multi-contextual and coarse-to-fine architecture. The proposed method is evaluated on several 3D MRI and CT datasets. The empirical results have shown the superiority of the proposed method compared with baseline models on high-resolution 3D volumes.

This paper has a clear problem formulation, and the proposed framework looks reasonable. For the experiments part, the datasets, baselines, evaluation metrics, and training details are empirically correct.

**Strengths:**

- The computational memory issue is critical for 3D super-resolution tasks. This paper aims to solve an important problem in the computer vision community by proposing a scalable transformer-based framework.

- The proposed method is technically sound. The combination of multi-contextual and coarse-to-fine modules is reasonable for developing an effective and scalable transformer-based architecture.

- This paper evaluates the model performance across five 3D datasets, which is comprehensive. The results are informative and support the main claims.

- This paper is generally well-written and well-organized.

**Weaknesses:**

- I am a bit unsure about the synthetic low-resolution generation process. On Page 5, the authors mentioned downsampling via linear interpolation. Why don’t the authors consider bicubic interpolation downsampling, as many super-resolution works do? The degradation process is critical since it will somehow misrepresent the real-world scenarios if not chosen well, especially when it comes to MRI reconstruction. It would be good to explore more degradation schemes and choose different degradation kernels for robustness.

- It would also be good to have more technical details on analyzing the hyperparameters to strengthen the paper. For example, when choosing window size/patch size for transformers, are they sensitive to selection? Also, using different parameters may pose challenges to the computational memory. It would be good to have practical guidance on choosing those parameters.

- Besides, I have several minor questions regarding this paper:

    - In Figure 5, the prediction of MTVNet produces somewhat blurry reconstruction results outside the blue box. Any illustrations on that?

    - For those MRI and CT datasets, are there any other evaluation metrics apart from the common PSNR and SSIM metrics? Maybe some statistical/frequency metrics?

    - The code link doesn’t work on my side.

**Final Justification:**

The authors have addressed my concerns. The proposed method is interesting, and the experiments/evaluations are solid. I recommend acceptance.

**Justification:**

This paper tackles a critical memory problem in 3D super-resolution. The proposed method is able to show superiority in high-resolution 3D reconstruction compared to baseline models. The training is done within 1 A100 GPU. The paper framework is generally easy to follow. However, there are some technical details that may require further clarification.

---

> ### Author Rebuttal · Authors · 2025-10-21
>
> We appreciate the insightful comments and constructive feedback from the reviewer. We would like to thank the reviewer for recognizing our work of addressing an important computational issue in volumetric super-resolution. We are also pleased that the reviewer finds our proposed multi-contextual modeling approach to be technically sound, finds our paper to be well-written and well-organized, and acknowledges the comprehensiveness of our experiments.
>
> Degradation approach: We appreciate the reviewer's comments on our choice of degradation. We agree with the reviewer's statement that bicubic interpolation is the most widely used degradation function for super-resolution in 2D. In volumetric super-resolution, however, linear interpolation is often used in conjunction with k-space degradation [1][2][3]. We chose to use linear interpolation for simplicity, yet we acknowledge that considering bicubic degradation would strengthen the experimental section of our paper. Still, we judge that adding additional degradation models would have minimal impact on our findings besides increasing statistical power, while requiring substantial effort.
>
> Hyperparameter analysis: We agree with the reviewer's comments that the inclusion of more detailed analysis of hyperparameters could be useful to some readers. We will extend the appendix with extra details on hyperparameter selection and their impact on computational memory usage in the final manuscript.
>
> LAM visualizations: We appreciate the reviewer's interest in our visualizations. In Fig. 5, the blue box encapsulates the super-resolution prediction area for all methods. Thus, we cannot illustrate any reconstruction results outside the blue box as they do not exist. The regions outside serve to provide our method with additional contextual information for the area inside the blue box. To showcase how our method leverages this extra context, we employ LAM. The red regions outside the blue box illustrate that the multi-contextual network stages of MTVNet enables our method to leverage broader image context to inform its prediction compared with other methods. We will improve the section on LAM visualizations to make the distinction between prediction area and contextual areas more clear in the final manuscript.
>
> Evaluation metrics: The evaluation metrics PSNR and SSIM are the most widely used metrics in field [4]. We acknowledge that the inclusion of domain-specific metrics could strengthen the findings of our paper. Yet, to our knowledge, such metrics have not gained broad usage within the super-resolution community, and the formulation of such metrics is beyond the scope of this work.
>
> Code link: We appreciate the reviewer's remark. We will fix the link and make our code available upon acceptance.
>
> [1] Y. Chen, et al. “Efficient and accurate MRI super-resolution using a generative adversarial network and 3D multi-level densely connected network”. In: International conference on medical image computing and computer assisted intervention. Springer. 2018.
>
> [2] C.-H. Pham, et al. “Brain MRI super-resolution using deep 3D convolutional networks”. In: 2017 IEEE 14th International Symposium on Biomedical Imaging (ISBI 2017). 2017, pp. 197–200.
>
> [3] C. Forigua, et al. “SuperFormer: Volumetric Transformer Architectures for MRI Super-Resolution”. In: Simulation and Synthesis in Medical Imaging. Ed. by C. Zhao, D. Svoboda, J. M. Wolterink, and M. Escobar. Cham: Springer International Publishing, 2022, pp. 132–141.
>
> [4] Z. Ji, et al. “Deep learning-based magnetic resonance image super-resolution: a survey”. In: Neural Computing and Applications (2024), pp. 1–28.

---

### Meta-Review · Area_Chair_preL · 2025-10-31

**Recommendation:** Accept (Poster)
**Confidence:** 3

**Metareview:**

This is a well-executed paper that advances the state-of-the-art in volumetric super-resolution with a practical and scalable approach. The comprehensive experiments, clear presentation, and empirical findings outweigh the minor limitations identified in reviews. The reviewers are all positive towards the paper and recommend acceptance. As suggested by two reviewers, I recommend accept as poster presentation.

---

### Decision · Program_Chairs · 2025-11-05

**Decision:**

Accept (Spotlight)

**Comment:**

We recommend an oral and a poster presentation given the AC and reviewers recommendations.

A spotlight presentation refers to a poster selected for an oral highlight but not designated as a full oral presentation per the AC’s recommendation.